# Uncertainty-Aware Pectoral Muscle Segmentation Based on Heteroscedastic Regression

**Paul Zech**[1,2] (ID)                                                    P.ZECH@SIEMENS-HEALTHINEERS.COM

**Christian Hümmer**[1]                                    CHRISTIAN.HUEMMER@SIEMENS-HEALTHINEERS.COM

**Christopher Syben**[1]                                 CHRISTOPHER.SYBEN@SIEMENS-HEALTHINEERS.COM

**Ludwig Ritschl**[1]                                         LUDWIG.RITSCHL@SIEMENS-HEALTHINEERS.COM

**Sebastian Stober**[2]                                                                  STOBER@OVGU.DE

[1] *Siemens Healthineers AG, Forchheim, Germany*

[2] *Otto-von-Guericke University, Magdeburg, Germany*

**Editors:** Accepted for publication at MIDL 2025

## Abstract

Accurate segmentation of the pectoral muscle is crucial for improving breast cancer diagnosis in mammograms. While modern deep learning models excel in segmentation, they often lack uncertainty quantification, which is essential for reliable clinical decisions. In this work, we propose a novel method for modeling uncertainty in pectoral muscle segmentation by combining the prediction of probabilistic heatmaps with heteroscedastic regression. For that, we investigate both an existing and a novel loss function derived from the heteroscedastic Laplace distribution, and show that our loss function is more robust for pectoral muscle segmentation in our setting. Further, we demonstrate that our method is capable of producing heatmaps with high-likelihood predictive distributions within a single model while outperforming an ensemble baseline in terms of accuracy.

**Keywords:** Pectoral muscle segmentation, uncertainty, heteroscedastic regression

## 1. Introduction

Breast cancer is one of the leading causes of mortality among all cancer types (Siegel et al., 2016). To improve early detection, digital mammography is the predominant method for screening patients due to its high cost-efficiency and effectiveness. In mammography image analysis, one important preprocessing step is the segmentation of the pectoral muscle (PM), for which various methods were introduced, ranging from traditional concepts that model the PM boundary (Ferrari et al., 2004) to modern deep learning based solutions that predict the PM region (Rampun et al., 2019), but they typically do not model uncertainty.

A common approach to modeling uncertainty is the assessment of variability in the predictive distribution generated by model ensembles. In the context of PM segmentation, these ensembles can be obtained using Monte Carlo dropout (Klanecek et al., 2023), different model states at various training stages (Tang et al., 2025) or training on different data subsets (Huemmer et al., 2024). However, ensembles primarily capture (epistemic) uncertainty in the model parameters rather than input-dependent heteroscedastic uncertainty (Cipolla et al., 2018; Chan et al., 2024), which is critical when the PM is partially undetectable due to dense glandular tissue and structural similarity to skin folds. In such ambiguous situations, modeling input-dependent uncertainty is crucial, as it is more informative than an overconfident yet inaccurate prediction. Hence, we propose a novel PM segmentation approach based on heteroscedastic uncertainty to capture local variations in noise and occlusions, requiring only a single model training.

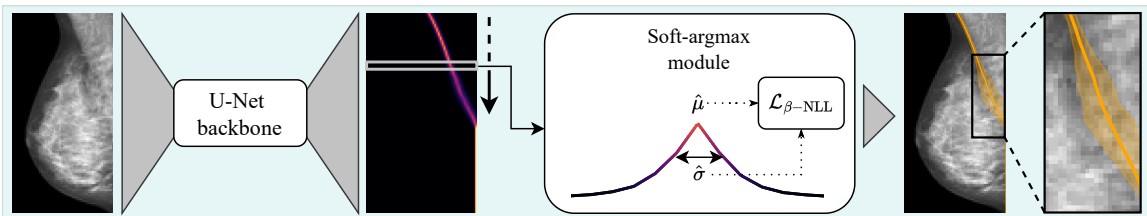

Figure 1: Illustration of our method, where the orange line shows the row-wise mean and the uncertainty band represents the row-wise variance of the predicted heatmap.

## 2. Methods

This paper builds on the work of Huemmer et al. (2024), who proposed to perform the PM segmentation as a regression task by leveraging a unique row-to-column index mapping to directly predict the PM boundary as a column-index (CI) vector. As a major difference, we employ an encoder-decoder architecture (U-Net) to perform heatmap regression through row-wise application of the soft-argmax operation (Luvizon et al., 2019), as shown in Figure 1. The integrated softmax operation enables a probabilistic interpretation and allows to compute relevant row-wise statistics from the heatmap

$$\hat{\mu}_i = \sum_j j \cdot h_{i,j}, \qquad \hat{\sigma}_i^2 = \sum_j h_{i,j} \cdot (j - \hat{\mu}_i)^2 \tag{1}$$

with mean $\hat{\mu}_i$ as PM boundary prediction, variance $\hat{\sigma}_i^2$ as measure for the uncertainty, and $h_{i,j}$ as value of the softmax-activated heatmap at row $i$ and column $j$. To train the uncertainty aware heatmap regression we utilize the $\beta$-NLL loss of Seitzer et al. (2022) which extends the standard heteroscedastic regression loss with a variance weighting for more stable convergence. It is derived from the negative log-likelihood (NLL) of the normal distribution and thus denoted as $\mathcal{L}_{\beta-\mathcal{N}\text{-NLL}}$. Initial experiments empirically revealed that squared error terms, such as the mean-squared-error (MSE) can lead to poor mean predictions compared to the mean-absolute-error (MAE). Hence, we propose a similar loss function $\mathcal{L}_{\beta-\text{Lap-NLL}}$ that is derived from the Laplace distribution and leads to an absolute difference term in the heteroscedastic loss:

$$\mathcal{L}_{\beta-\text{Lap-NLL}} := \frac{1}{N} \sum_{i=1}^{N} \left\lfloor \hat{b}_i^{\beta} \right\rfloor \left( \frac{|y_i - \hat{\mu}_i|}{\hat{b}_i} + \log(2\hat{b}_i) \right) \quad \text{with } \hat{b}_i = \sqrt{\frac{\hat{\sigma}_i^2}{2}}. \tag{2}$$

Here, $\lfloor . \rfloor$ denotes a gradient-stop operation and $y_i$ refers to the target at row $i$. Equivalent to $\mathcal{L}_{\beta-\mathcal{N}\text{-NLL}}$ (Seitzer et al., 2022), $\hat{b}_i^{\beta}$ is multiplied to the NLL to interpolate between uniform sample weighting and the NLL loss by selecting $\beta$ between 0 and 1.

Furthermore, we extend the loss function regularization in (2), inspired by the static variance regularization of Nibali et al. (2018), by using

$$\mathcal{L}_{\text{reg}} = \frac{1}{N} \sum_{i=1}^{N} \mathcal{D}_{\text{JS}}(\mathbf{h}_i \| \mathcal{Q}_i) \quad \text{with} \quad \mathcal{Q}_i = \begin{cases} \mathcal{N}(\hat{\mu}_i, \hat{\sigma}_i^2) & \text{(Gaussian)} \\ \text{Lap}(\hat{\mu}_i, \hat{b}_i) & \text{(Laplace)} \end{cases} \tag{3}$$

where $\mathcal{D}_{\text{JS}}(\mathbf{h}_i \| \mathcal{Q}_i)$ denotes the Jensen-Shannon divergence between the softmax-activated heatmap $\mathbf{h}_i$ (row index $i$) and a template distribution $\mathcal{Q}_i$, constructed from the predicted heatmaps' statistics in (1). The regularization term of (3) is weighted by a constant factor $\lambda = 100$ and added to the NLL loss with the goal to produce meaningful distributions that align with the probabilistic model of the underlying loss function.

## 3. Experiments and results

We evaluate our method on 8861 mediolateral oblique (MLO) mammograms (both left and right laterality) from the MBTST dataset (Dahlblom et al., 2019). Ground truth labels were provided by clinical experts. For independent testing, 15% of the images were extracted while keeping patient boundaries. Mammograms were resized to $128 \times 128$, normalized to range of $[0, 1]$, and edge-padded by $1/4$ of their spatial dimensions for boundary context. Left-lateral images were horizontally flipped for consistent right-laterality. We trained our method using the $\mathcal{L}_{\beta-\mathcal{N}\text{-NLL}}$ and $\mathcal{L}_{\beta-\text{Lap-NLL}}$ with various $\beta$ parametrizations, and compared it against the CI vector regression ensemble of DenseNets by Huemmer et al. (2024). All architectures were adapted to have $\sim 450\text{k}$ parameters. The results shown in Table 1 are averaged over five runs with different data splits, maintaining the same test set. Aligning with prior assumptions, our method using $\mathcal{L}_{\beta-\text{Lap-NLL}}$ produces more accurate mean fits, particularly for higher $\beta$-values, where $\beta$ controls the tradeoff between accuracy and log-likelihood (LL). Additionally, it outperforms the ensemble baseline in terms of accuracy while providing high-likelihood predictive distributions within a single model. Upon visual inspection of the test set, we observed an increase in row-wise variance in regions where the muscle outline is obscured, indicating valuable information about the models' uncertainty in difficult situations. This is further supported by the Pearson coefficient, which measures image-wise correlation between absolute error and variance.

Table 1: Comparison of our method against an ensemble baseline.

| Method | Loss | $\beta$ | MAE ↓ | RMSE ↓ | Pearson ↑ | LL ↑ |
|---|---|---|---|---|---|---|
| Ensemble | SAE | $-$ | $0.53 \pm 0.00$ | $0.98 \pm 0.01$ | $0.70 \pm 0.01$ | $-87.63 \pm 10.60$ |
| U-Net (Ours) | $\mathcal{L}_{\beta\text{-}\mathcal{N}\text{-NLL}}$ $+ \lambda \mathcal{L}_{\text{reg}}$ | 0.0 | $0.52 \pm 0.02$ | $1.09 \pm 0.06$ | $0.69 \pm 0.00$ | $51.83 \pm 16.89$ |
| | | 0.5 | $0.48 \pm 0.01$ | $1.01 \pm 0.02$ | $0.69 \pm 0.02$ | $-20.13 \pm 55.95$ |
| | | 1.0 | $0.53 \pm 0.04$ | $1.09 \pm 0.05$ | $0.68 \pm 0.03$ | $-75.79 \pm 48.96$ |
| | $\mathcal{L}_{\beta\text{-Lap-NLL}}$ $+ \lambda \mathcal{L}_{\text{reg}}$ | 0.0 | $0.51 \pm 0.02$ | $1.12 \pm 0.03$ | $0.72 \pm 0.01$ | $111.96 \pm 11.94$ |
| | | 0.5 | $0.47 \pm 0.00$ | $1.01 \pm 0.02$ | $0.71 \pm 0.00$ | $104.71 \pm 6.01$ |
| | | 1.0 | $0.46 \pm 0.01$ | $0.98 \pm 0.01$ | $0.72 \pm 0.00$ | $87.18 \pm 10.64$ |

## 4. Conclusion

In conclusion, we introduce a novel approach to PM segmentation that models input-dependent uncertainty. Preliminary results are promising and suggest to further validate the methods ability to provide meaningful and reliable uncertainty measures.

**Disclaimer:** The presented methods in this paper are not commercially available and their future availability cannot be guaranteed.

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
