# OpenReview forum: "Uncertainty-Aware Pectoral Muscle Segmentation Based on Heteroscedastic Regression"
_MIDL.io/2025/Short_Papers — MIDL 2025 - Short Papers_

### Official Review · Reviewer_c4ny · 2025-04-28

**Rating:** 3
**Confidence:** 5

**Summary:**

This study focuses on the segmentation of the pectoral muscle based on a row-to-column index regression strategy. The originality of the method lies in the integration of uncertainty to improve the model performance. Experiments were conducted using a dataset composed of 8,861 mammograms. Results show small but consistent improvements with the proposed solution, particularly for large beta values.

**Strengths:**

The strengths of this work are:
1) the relevance of the topic of this study, any improvement/innovation of which could have a major impact in our field
2) the idea of integrating uncertainty through a heteroscedastic regression loss
3) the size of the dataset used for the experiments (i.e. 8,861 mammograms)

**Weaknesses:**

The main weaknesses of this article concern:
1) Lack of justification for the different choices that have been made: full segmentation vs. row-to-column index regression, the interest in using the Beta-NLL loss, MSE vs. MAE, and the role of the regularization term
2) No ablation studies on the interest of adding the regularization term were performed.
3) Only mean values are reported in the results table, preventing an analysis of the significant improvements brought by the solution

---

### Decision · Program_Chairs · 2025-05-01

Accept